# Photochemical Efficiency and Oxidative Metabolism of Tree Species during Acclimation to High and Low Irradiance

**DOI:** 10.3390/plants9081047

**Published:** 2020-08-17

**Authors:** Adamir da Rocha Nina Junior, Jair Max Furtunato Maia, Samuel Cordeiro Vitor Martins, José Francisco de Carvalho Gonçalves

**Affiliations:** 1Federal Institute of Education, Science and Technology of Amazonas (IFAM)–Campus Humaitá, BR230 Highway, KM 07, Humaitá-AM 69.800-000, Brazil; adamir.nina@ifam.edu.br; 2University of State of Amazonas (UEA), Av. Djalma Batista, 2470, Manaus-AM 69.050-010, Brazil; jmaia@uea.edu.br; 3Department of Plant Biology, Federal University of Viçosa (UFV), Av. PH Rolfs, Viçosa-MG 36570-900, Brazil; samuel.martins@ufv.br; 4National Institute for Amazonian Research (INPA), Laboratory of Plant Physiology and Biochemistry, Av. André Araújo, 2936, Aleixo, Manaus-AM 69.011-970, Brazil

**Keywords:** chlorophyll fluorescence, nutritional status, photoinhibition effects, reactive oxygen species, successional groups, tropical tree species

## Abstract

The balance between efficiency of absorption and use of light energy is fundamental for plant metabolism and to avoid photoinhibition. Here, we investigated the effects of light environments on the photosynthetic apparatus of tropical tree species of three successional groups (pioneer, mid-, and late successional) subjected to different light conditions: full sunlight (FS), moderate shade (MS), and deep shade (DS). Twenty-nine ecophysiological parameters were correlated with each other. The pioneer species exhibited better photochemical performance and a more efficient antioxidant enzymatic system in comparison with the other successional groups. Plants in FS showed higher intensity of lipid peroxidation, with superoxide dismutase having a prominent role in the antioxidant system. At lower irradiance the enzymatic activity was reduced, and the photochemical efficiency was the preferred way to reduce oxidative damages. P was highly related to photochemical yield, and the N modulation amplified the light harvesting complex in DS to the detriment of the antioxidant system. Despite evidence of cell damage, most species exhibited the ability to adjust to high irradiance. Contrary to expectations, *Hymenea courbaril* (late-successional) exhibited higher plasticity to fluorescence, nutritional, and antioxidant parameters. Only *Carapa guianensis* (late-successional) displayed photoinhibitory damage in FS, and *Ochroma pyramidale* (pioneer) did not survive in DS, suggesting that acclimation to shade is more challenging than to high irradiance.

## 1. Introduction

As a primary source of energy, light influences physiological processes, being essential for the flow of energy in biological systems and highly determinant for plant growth and development. As the light supply can be very heterogeneous, the plants, more specifically their leaves, must exhibit great flexibility of physiological responses to cope with light fluctuations. Given the spatial and seasonal changes in the availability of primary resources, plants with low physiological flexibility would likely be constrained in their ability to survive in changing environments [1,2,3].

The ability of a genotype to express different phenotypes in response to environmental conditions is known as phenotypic plasticity, which is a key feature for acclimation [4]. When exposed to changes in light environment, plants acclimate to the new condition through adjustments in leaf characteristics and physiological functions. In this sense, different acclimation responses to environmental stress are expected among plants depending on their age, phenology, successional group [5], besides, of course, the influence of other abiotic (such as soil-water and nutrients availability) and biotic factors (for example, competitors, predators, and phytopathogen). Several studies suggest that pioneer species are more plastic than late-successional species; however, increasing evidence indicates that this capacity to modulate the photosynthetic apparatus is not necessarily related to ecological groups [6]. It seems that, independent of the successional group, these modifications occur to maximize the light use efficiency and minimize the occurrence of photoinhibition, i.e., the decline of photosynthetic activity when light energy is absorbed beyond the photosynthetic capacity of the plant [7,8,9].

One way to reduce the effects of excessive irradiance is to dissipate the excess energy in nonphotochemical processes, such as dissipation in the form of heat, via the xanthophyll cycle, and in the form of fluorescence [10]. Many studies have been carried out using measurements of chlorophyll *a* fluorescence to evaluate photoinhibitory damage caused by excess energy and action of reactive oxygen species (ROS), which are known to cause cellular damage [11]. The ROS are inevitable byproducts of a physiological process that involves oxidation-reduction reactions, such as the electron transport chains in chloroplast and mitochondria. If ROS are not scavenged, they can lead to oxidation of important cellular components, such as cell membranes, lipids, and even the genetic material of the cell. To avoid oxidative damage, plants have enzymatic and nonenzymatic systems for removal of ROS, which may be directly related to the tolerance of plants to the stress situation [12,13]. In addition, the antioxidant enzyme SOD (superoxide dismutase) is considered the first line of defense in the fight against ROS, transmuting O_2_^●−^ to form H_2_O_2_. In addition, the CAT (catalase), APX (ascorbate peroxidase), and POX (peroxidase) enzymes complement the ROS elimination process by transforming H_2_O_2_ into water and molecular oxygen [14,15].

All plants grow under a dynamic balance among light, nutrients, and other primary factors. However, in conditions where light is limiting, the increase of nutrient concentrations in leaves plays a major role in the use of light energy. This is due to the requirements for N, P, and micronutrients for the synthesis of the photosynthetic apparatus [2,16]. On the other hand, it is believed that, in plants subjected to high irradiance, an adequate mineral nutrition can improve the defense mechanisms against high light stress [17,18].

The dynamic of the light regime imposes a series of difficulties for the plant survival in natural or crop environments. In this scenario, the investigation of the physiological performance of young plants concerning the incidence of light is justified, given not only the heterogeneity with which this resource presents itself in a natural environment but also because of its strong influence on plant metabolism. Here, based on the previous considerations, we hypothesized that tropical tree species grown under low light exhibit less ability to overcome stress when compared to trees grown under high light conditions, regardless of their successional group. In this context, the objective of this study was to investigate the photochemical efficiency and the antioxidative system of tropical tree species grown under different light environments to test if there is a precise correspondence between photochemical performance and classification of successional stages in species of the Amazon forest.

## 2. Material and Methods

### 2.1. Plant Material and Growth Conditions

Saplings of six native Amazonian tree species belonging to three distinct successional groups (Table 1) were taken to the National Institute for Amazonian Research (INPA, Manaus, Amazonas - Brazil) and cultivated in the nursery. When they reached nine months of age, they were transplanted into plastic pots containing 12 L of substrate, a dystrophic Red Yellow Latosol (Oxisol) collected in the native forest with organic matter (see Appendix A). At 12–14 months of age, one part of the group was transferred to two irradiance environments and one part was kept in the nursery (4 individuals per species in each treatment). Incident photosynthetic radiation was monitored with a line quantum sensor (model LI-191, LI-COR Inc., Lincoln, Nebraska, USA) for 7 sunny days.

The light treatments (Table 2) consisted of full sunlight (FS) (100% of solar irradiance, simulating a forest clearing); artificial moderate shade (MS), provided by shade cloths reducing direct, incident solar radiation (simulating an understory light environment with partial canopy openness), and natural deep shade (DS), with natural shade provided by adult tree canopies (simulating an understory light environment). The plants were subjected to these treatments for 180 days.

### 2.2. Chloroplastidic Pigments Content

Fully expanded and healthy leaves were collected, covered with aluminum paper, immediately frozen in liquid nitrogen, and stored at −80 °C until the moment of analysis.

The analysis was performed using 0.1 g of fresh material ground in 10 mL of 80% acetone with magnesium carbonate (MgCO_3_) and 10 mL of 100% acetone added immediately following the initial grinding step. The suspension was filtered, and the absorbance was read at 663 nm (Chl *a*), 645 nm (Chl *b*), and 480 nm (Car) using a spectrophotometer (Ultrospec 2100 pro UV/visible, Amersham Biosciences, Cambridge, UK) [25]. The chlorophylls *a* (Chl *a*) and *b* (Chl *b*) and carotenoids (Car) content were calculated using equations described by [26], as follows:$$\mathrm{Chl}\;a\;\left({µ\mathrm{mol}\;g}^{-1}{\;\mathrm{or}\;µ\mathrm{mol}\;m}^{-2}\right)=\frac{\left(12.7\;\times \;A663\;–\;2.69\;\times \;A645\right)\;\times \;1.119\;\times \;V}{1000\;\times \;\mathrm{area}\;\mathrm{unit}\;\left(m^{2}\right)\;\mathrm{or}\;\mathrm{fresh}\;\mathrm{mass}\;\left(g\right)}$$
$$\mathrm{Chl}\;b\;\left({µ\mathrm{mol}\;g}^{-1}{\;\mathrm{or}\;µ\mathrm{mol}\;m}^{-2}\right)=\frac{\left(22.9\;\times \;A645\;–\;4.68\;\times \;A663\right)\;\times \;1.102\;\times \;V}{1000\;\times \;\mathrm{area}\;\mathrm{unit}\;\left(m^{2}\right)\;\mathrm{or}\;\mathrm{fresh}\;\mathrm{mass}\;\left(g\right)}$$
$$\mathrm{Car}\;\left({µ\mathrm{mol}\;g}^{-1}{\;\mathrm{or}\;µ\mathrm{mol}\;m}^{-2}\right)=\frac{\left(A480\;+\;0.114\times \;A663\;–\;0.638\times \;A645\right)\;\times \;V\;\times \;1000}{112.0\;\times \;\mathrm{area}\;\mathrm{unit}\;\left(m^{2}\right)\;\mathrm{or}\;\mathrm{fresh}\;\mathrm{mass}\;\left(g\right)}$$

A (given wavelength) represents the absorbance at the indicated wavelength, V is the final volume of the chlorophyll–acetone extract (mL). The total chlorophyll (Chl_tot_) content is the sum of Chl *a* and Chl *b*. For their ecophysiological implications, we calculated ratios Chl *a*/Chl *b* and total Chl_tot_/Car content.

### 2.3. Chlorophyll Fluorescence Parameters

Leaf gas exchange and chlorophyll *a* fluorescence were measured simultaneously using an open-flow infrared gas exchange analyzer system (LI-6400XT, Li-Cor, Lincoln, NE, USA) equipped with a leaf chamber fluorometer (Li-Cor 6400-40 chamber with light source coupled and 2 cm^2^ of measurement area). The measures were taken between 7:00 a.m. and 1:00 p.m.

Fully expanded and healthy leaves were sampled and acclimated to darkness for 30 min and a weak modulated measuring beam (0.03 μmol m^−2^ s^−1^) was applied to obtain the minimal fluorescence (F_0_). The maximum fluorescence emissions (F_m_) were measured after applying a saturating white light pulse of 8000 μmol m^−2^ s^−1^ for 0.8 s.

The leaf was enclosed in the gas exchange system and left under baseline conditions until net assimilation (A), stomatal conductance (g_s_), and internal CO_2_ concentration (C_i_) stabilized. The baseline conditions inside the leaf cuvette included CO_2_ concentration (Ca) 400 μmol mol^−1^, relative humidity around 60%, 30 °C temperature, and PPDF of 1500 μmol m^−2^ s^−1^. After the acclimation period, the fluorescence signals F*_s_* (steady-state fluorescence under actinic illumination of 1500 μmol m^−2^ s^−1^), F*_m_*′ (maximum fluorescence during a light-saturating pulse of 8000 μmol m^−2^ s^−1^), and F*_0′_* (light-adapted minimal fluorescence, obtained using a weak far-red illumination) were measured simultaneously with photosynthetic responses to irradiance (A/PPDF curves) in 14 photosynthetic photon flux density (PPFD) levels between 0 and 2000 μmol m^−2^ s^−1^ in decreasing order (2000, 1500, 1000, 750, 500, 300, 200, 150, 100, 75, 50, 25, 10, and 0 μmol m^−2^ s^−1^).

The calculated parameters are show in Table 3.

Nonlinear regression models were fitted to describe the variations in photochemical and nonphotochemical yields with PPFD for each sapling [35].

### 2.4. Extraction of Soluble Protein and Measurement of Enzyme Activities

Leaves in the same conditions as those collected for the pigment analysis were collected at 1:00 p.m. and immediately frozen in liquid nitrogen. For the analysis of enzyme activity, 200 mg of leaf samples were ground to a fine powder and homogenized in 100 mM potassium phosphate buffer (pH 6.8) containing 1mM EDTA, PMSF 1 mM, and 1% (w/v) soluble PVPP. The homogenate was centrifuged at 15,000× *g* for 20 min. The supernatant was collected, and aliquots were used for enzyme analysis. All extraction procedures were performed at 4 °C. 

The activity of Catalase (CAT; EC 1.11.1.6) was measured spectrophotometrically at 270 nm by determining the rate of H_2_O_2_ conversion to O_2_ [36].The activity of ascorbate peroxidase (APX; EC 1.11.1.11) was measured as a decrease in absorbance at 290 nm, which results from ascorbate oxidation [37]. The activity of peroxidases (POX; EC 1.11.1.7) was measured as a decrease in absorbance at 420 nm, which results from purpurogallin formation [38] The superoxide dismutase (SOD; EC 1.15.1.1) was measured spectrophotometrically at 560 nm by detecting the inhibition of nitroblue tetrazolium (NBT) reduction by SOD [39]. One unit of SOD was defined as the amount needed for 50% inhibition of NBT reduction [40]. Total soluble protein was determined according to the Bradford method [41], using bovine serum albumin (BSA) as the standard.

### 2.5. Lipid Peroxidation (Determination of Malonaldehyde Content)

The lipid peroxidation was estimated by formation of thiobarbituric acid (TBA) reactive substances [42]. Absorbance was measured at 535 and 600 nm, and malonaldehyde (MDA) concentrations were calculated using an extinction coefficient of 155 mM^−1^ cm^−1^ [43].

### 2.6. Leaf Phenols Compounds

Total phenolic compounds were determined by [44] and using the Folin–Ciocalteu reagent. The absorbance was measured at 725 nm and utilized tannic acid in the calibration curves.

### 2.7. Leaf-Nutrient Content

The leaves samples were dried in an oven at 65 °C to mass constant. The total N was determined by the Kjeldahl method. Macronutrients (Ca, Mg, P, and K) and micronutrients (Fe, Zn, Cu, and Mn) were extracted by digestion with 3:1 nitric–perchloric solution, concentrations of these nutrients were determined by atomic absorption spectrometry (Perkin-Elmer 1100B, Uberlingen, Germany) [45], and P was determined by spectrophotometry at 725 nm [46].

### 2.8. Plasticity Index

The plasticity index (PI) for the fluorescence, nutritional, and antioxidant parameters were calculated for each species using the maximum and minimum values observed for each variable, as described in [47], where:$$PI=\frac{Maximum\;mean\;value-Minimum\;mean\;value}{Maximum\;mean\;value}$$

The PI values range from 0 (no plasticity) to 1 (maximal plasticity).

### 2.9. Experimental Design and Statistical Analysis

The experimental design was completely randomized factorial (6X3) with six species in three light treatments. After the assumptions of normality and homoscedasticity were checked, to analyze the differences between species and treatments, the appropriate analyses of variance and the averages were performed using the Tukey post hoc test. When appropriate, relationships between variables were tested by regression equations, using the following criteria for adjustment: (1) significance of the adjusted regression; (2) significance of its coefficients, and (3) higher coefficient of determination. The interrelationships among functional trait variables were assessed using the principal components analysis (PCA) ordination method.

All the tests and statistical analyses were performed with Statistica 8.0 (Statsoft Inc., Tulsa, OK, USA, 2007), SPSS 23 (IBM Corp. 2015) and SigmaPlot 11 (Systat software, 2008). To determine the relationship between the different variable parameters, we calculated the Pearson’s correlation coefficients. The correlation matrixes were prepared by R program (http://www.r-project.org). The values of the correlation coefficient varied between +1 and −1. When the value is around +1 or −1, it indicates a close positive or negative relationship between the variables, respectively. As the correlation coefficient value approximates zero, the relationship between the two variables becomes weaker.

## 3. Results

### 3.1. Chloroplastidic Pigments Content

Plants in DS exhibited higher contents of Chl *a* and *b*, on an area and mass basis, than FS plants. In MS only *H. courbaril* and *T. serratifolia* matched the contents found in DS, conferring them greater photosynthetic advantage when compared with others in this environment. This was repeated for the *Car* and Chl_tot_ concentrations in the area base, but, in the mass base, both total pigments and individual concentrations were higher in DS than in the other environments (Appendix A).

The Chl *a*: Chl *b* ratio decreased with decreasing irradiance (higher in FS), while Chl_tot_: *Car* showed opposite behavior (higher in DS). It is important to note that both are important physiological indicators of the adaptation of plants to light environments.

### 3.2. Chlorophyll Fluorescence Parameters

Concerning the chlorophyll *a* fluorescence, only *C. guianensis* showed an F_v_/F_m_ ratio lower than 0.75, and these values were found in FS, denoting higher susceptibility of these species to photoinhibition when under high irradiance. Except for these two, despite small differences between absolute values, other studied plants showed no signs of exacerbated photochemical limitations (Figure 1A).

In general, the ETR was higher in FS than in other environments. Under saturating light (1500 µmol m^−2^ s^−1^), *B. grossularioides* and *O. pyramidale* exhibited the highest values, the first being 1.8 and 2.1 times greater than MS and DS and the second 1.5 times greater than MS, respectively (Figure 1B). All individuals of *O. pyramidale* (pioneer) in DS were already dead at 97 days of experiment. In MS, the highest values observed were for *H. courbaril*, followed by *T. serratifolia*. *B. grossularioides* and *H. brasiliensis* displayed higher ETR in DS, however, while the first showed high performance in FS, the second had little difference between the environments.

Only for *C. guianensis* the fraction of electrons destined for photorespiration was greater in full sunlight than in the other environments, denoting the lower efficiency of this species under high irradiance (Figure 1C). In absolute terms, the fraction of electrons destined for carboxylation (ETR_C_) was higher in FS plants than MS for *H. brasiliensis* (+11.1%), *B. grossularioides* (+41.2%), and *O. pyramidale* (+28.3%). The late-successional *H. courbaril* and *C. guianensis* showed higher ETR_C_ in DS than FS (+19.5% and +11.6%, respectively). On the other hand, FS plants proportionally exhibited the largest electron fractions for oxygenation (ETR_O_), denoting the highest photorespiratory cost of the plants in this environment (Appendix A).

Higher F_v_′/F_m_′ values were observed for DS plants and lower FS plants (Figure 1D). In FS, the highest value was observed in *H. courbaril* and the lowest in *C. guianensis* (−25.5%). In MS, *H. courbaril* values were also higher, but this time the species with lower values was *B. grossularioides* (−33.6%). *B. grossularioides* had the highest yields in DS, while *H. brasiliensis* had the lowest (−19.3%).

In contrast to F_v_′/F_m_′, photochemical quenching (qL) was higher for all species in FS, with emphasis on *T. serratifolia* and *B. grossularioides*, which were 9.4 and 5.1 times higher in qL than the plants in DS (Figure 1E). With a higher necessity for energy dissipation in FS, NPQ followed the same behavior as qL, with the same species being 2.1 and 1.8 times higher in NPQ than that observed in for plants in DS. Both were followed by *O. pyramidale*, which was 1.2 times higher in FS than those in MS (Figure 1F).

The ɸPSII was greater in the pioneers than in the other species in FS. In the other environments (MS and DS), the opposite was observed (Figure 1G). *C. guianensis* was the species that exhibited lower ɸPSII in all environments.

Under normal conditions (21% O_2_), the PPDF in which ɸNPQ exceeds ɸPSII (ɸPSII=ɸNPQ) was higher in FS plants than in DS plants, except for *C. guianensis* and *H. brasiliensis*, which did not differ between environments (Figure 1H). In DS the species did not differ in this parameter, but in other environments, *C. guianensis* exhibited the lowest ɸPSII=ɸNPQ values. In FS, the highest ɸPSII=ɸNPQ observed value was for *B. grossularioides* (about 564 µmol m^−2^ s^−1^) and in MS for *H. courbaril* (about 517 µmol m^−2^ s^−1^) (Figure 1H). In general, under conditions in which photorespiration is theoretically suppressed (1% O_2_), ɸNPQ exceeds ɸPSII at lower irradiances in FS, denoting the importance of this process in photochemical quenching. Depending on the extent in which the irradiance decreases between environments, ɸPSII increases, decreasing the participation of ɸNPQ (Appendix A).

### 3.3. Leaf-Nutrient Content

The N and Fe contents were strongly influenced by the environment, increasing with less irradiance. In all environments, the highest values observed were for intermediate species *H. brasiliensis* and *T. serratifolia* (Appendix A). Late-successional species exhibited the lowest P content in all environments, with pioneers exhibiting almost twice in FS, as observed for K. For Mg, there was little variation between species and environments. *H. courbaril* and *T. serratifolia* exhibited the highest Mn content in all environments, with the first showing the highest values in FS and the second in DS, while *H. brasiliensis* did not differ among treatments. The lowest Zn levels were obtained in FS, except for *C. guianensis*, which did not differ among environments (Appendix A).

### 3.4. Enzymatic Activities, Leaf Phenolic Compounds, and Lipid Peroxidation 

The major activities recorded for CAT were in *H. brasiliensis* in FS and MS. Only *C. guianensis* did not differ between treatments. The general trend of CAT activity suggests that higher light conditions induce higher activity of this enzyme, especially in nonpioneer species (Appendix A). APX, on the contrary, displayed higher activity for *H. courbaril* and *C. guianensis* in DS than in FS (+99% and +84.9%, respectively). Pioneer species exhibited less expressive results in all environments for this enzyme (Appendix A).

The late-successionals *H. courbaril* and *C. guianensis* also showed higher POX activity in DS than when in FS, as well as the pioneer *B. grossularioides* (+97.4%, +60.4%, and +50.8%, respectively). Among the intermediates, *H. brasiliensis* did not differ between treatments and *T. serratifolia* showed lower activity in DS (−31.1%). For the pioneer *O. pyramidale*, the activity in MS was 56.8% higher than in FS (Appendix A).

As POX, the SOD activity in DS was also higher in *H. courbaril*, *C. guianensis*, and *B. grossularioides* than FS (5.8, 2.9 and 2.8 times greater). *H. brasiliensis* had higher SOD activity in MS 1.5 times higher in FS and 18 times higher in DS, whereas *T. serratifolia* in FS showed activity 2.4 and 1.4 higher than when in MS and DS, respectively. *O. pyramidale* did not differ between treatments (Appendix A).

Phenolic compounds also have an antioxidant effect and, in general, the nonpioneer species exhibited higher total phenol content in FS than pioneer, late-successional species *H. courbaril* and *C. guianensis*; about 1.8 times higher than the others. For the pioneers, the highest values observed were in MS. *B. grossularioides* exhibited higher phenol content in DS, and the lowest content in this environment was found in *H. brasiliensis* and *T. serratifolia* (−72.6%) (Appendix A).

The test for lipid oxidation was evaluated by thiobarbituric acid reactive substances (TBARS), and the results were lower in DS than FS, except for *C. guianensis*, which did not differ between these environments. *O. pyramidale* exhibited the lowest content of TBARS in FS, and the highest values were observed for *B. grossularioides* and *H. courbaril*. In DS and MS, *B. grossularioides* had lower TBARS, while *C. guianensis* in MS had higher TBARS than the others in this environment (Appendix A).

### 3.5. Plasticity Index

The plasticity indices for the species under different light conditions are displayed in Table 4. The most plastic characteristic was qL (0.86), and the least plastic was F_V_/F_M_ (0.21), both fluorescence parameters. Analyzing the results by parameter, the range of variation reduced considerably, without difference between the fluorescence and nutrient parameters. The antioxidant parameter exhibited greater plasticity and is the only one in which the species differ, with *H. courbaril* (late-successional) exhibiting the largest PI and *O. pyramidale* (pioneer) the smallest.

Evaluating the individual behavior of the species among all parameters, *O. pyramidale* (pioneer) showed the lowest index of plasticity followed by *C. guianensis* (late-successional) and *T. serratifolia* (intermediate), and the most plastic species was *H. courbaril* (late-successional).

### 3.6. Relationships between Light Capture and Use Parameters, Antioxidant Activity, and Leaf-Nutrient Content

In general, the photochemical parameters in FS were more related to the content of P than in other environments. It is important to highlight the photoprotective role of CAR (*r* = −0.73) and the SOD (*r* = 0.74) activity acting together to reduce oxidative damage (TBARS). The strong relationship between N and enzymatic activity and especially of Fe with SOD activity (r = 0.64) is also observed (Figure 2).

Interspecific responses to light intensity in each environment were evaluated using PCA. In FS, the PCA explained 45.44% of the variation of the data. Axis 1 of the PCA explained 23.79% of the data (0.6 < *r* < 0.9) (strong and positive). Strong and negative correlations were observed in this axis with ɸPSII, ETR, and the leaf content of Mg and P. Axis 2 explained 20.75% of data variation, and the very strong positive correlations (*r* > 0.9) were with SOD activity and strong correlations with phenolic content, Chl*a*, Chl*b*, CAR, and Chl_tot_ on a mass basis. Strong negative correlations were between axis 2 and N foliar concentration (Figure 2).

In contrast to FS, in MS there was lower participation of P in photochemical processes, especially in ɸPSII and ETR. The activity of SOD and the phenolic compounds performed discretely in the decrease of the damage (Figure 3).

The PCA for MS explained 51.08% of the variation of the data, and in axis 1 is contained 30.48% of the power of explanation. With this axis the content of pigments, F_v_′/F_m_′, ɸPSII, ETR, N and Mn content, and ɸPSII=ɸNPQ were strongly correlated. In this axis, the strong and negative correlations were with the content of phenols and ɸNPQ. Axis 2 explained 20.60% of the data and was strongly related to APX and POX activity, Chl*a*: Chl*b*, and Ca content. A negative correlation was observed with qL (Figure 3). 

In DS, there was a negative correlation between N content and enzymatic activity, as opposed to environments with higher irradiance, evidencing the investment of N and P foliar in light capture. Therefore, the antioxidative system had lower participation in protection against oxidative damage (TBARS), that was reduced with increases in ɸPSII, ETR, and also by the maintenance of higher PSII yield at higher irradiance (ɸPSII = ɸNPQ) (Figure 4).

The PCA for DS explained 53.30% of the variation of the data. Axis 1 contains 32.58% of the explanation and was strongly and positively related to the activity of CAT, TBARS, Chl*a*: Chl*b*, and Chl_tot_: CAR. Negative correlations occurred between axis 1 and pigment content, ɸPSII, ETR, content of P and Zn, and ɸPSII = ɸNPQ. With axis 2, containing 20.72% of the data variation, the positive correlations were with F_v_′/F_m_′, ɸNO, and the content of K and Fe. Negative correlations were between NPQ and ɸNPQ (Figure 4).

## 4. Discussion

### 4.1. Chloroplastidic Pigment Content

Photosynthetic pigments are constantly degraded and synthesized in the presence of light, but, under conditions of high irradiance, the degradation occurs at a higher rate than the synthesis, justifying the lower concentrations of pigments found in FS [5,48,49,50]. In addition, the reduction of pigment content suggests a smaller light harvest complex, which may prevent the absorption of excess light energy and avoid oxidative damages. On the other hand, in DS, there was an increase in pigment content to increase the light interception surface to compensate for the low availability of the resource in this environment.

The higher content of Chl *b* in DS favors the absorption of light at wavelengths higher than those of Chl *a*, which are more abundant because of the diffuse radiation that reaches the understory. Therefore, the Chl *a*: Chl *b* ratio is an important indicator of the plant adaptability to different light environments [51]. As expected, Chl *a*: Chl *b* ratio was higher in DS than in other environments.

Since there is a need to maintain a certain pigment stoichiometry, the DS plants also exhibited a higher *Car* concentration. Despite the higher content in DS, the Chl*_tot_*: CAR ratio was higher in FS, demonstrating the higher proportion of carotenoids in relation to chlorophylls in this environment given its important photoprotective role. The increase of these pigments in the xanthophyll cycle is fundamental for the dissipation of excess energy under high irradiance conditions [52].

### 4.2. Chlorophyll Fluorescence Parameters

Among the species studied, only *C. guianensis* exhibited photoinhibition effects, evidenced by the lower F_v_/F_m_ ratio (0.67). Values close to 0.80 indicate maximum efficiency in the use of energy in the photochemical process, while values below 0.75 indicate a stress situation in which there is a reduction of the plant photosynthetic potential [53,54]. Other studies have reported a reduction of the F_v_/F_m_ ratio in late-successional species, including *C. guianensis*, when submitted to high irradiance [55,56,57]. 

The highest and lowest ETR values were observed in FS and DS conditions, respectively, which was compatible with the energy availability in the environment. In plants under stress, when the electron flux in the photochemical stage is intense, alternative routes to carboxylation can be activated to dissipate excess energy preventing photochemical damage, and photorespiration is one of the main ways of consuming excess energy [58]. The highest total ETR in FS was accompanied by a higher flow of electrons destined for photorespiration in this environment, notably for *C. guianensis* and *T. serratifolia*, which exhibited 10% and 7% higher FS photorespiratory costs when compared to plants in DS. 

The F_v_′/F_m_′, qL, and ETR were strongly influenced by the light environment. These parameters, which represent the portion of the excitation energy captured by the open PSII reaction centers and the proportion of electrons used in the photochemical phase, indicated that plants were able to use the ambient radiation, although damage occurred to *C. guianensis*. It can be seen that, despite FS, when the plants had a lower portion of open PSII, they exhibit the higher qL under saturating light, resulting in greater energy available for the Calvin cycle. This increase in qL may be a consequence of the higher rates of electron transport around the photosystems in FS.

Only *T. serratifolia* and *B. grossularioides* exhibited differences in NPQ between FS and DS under saturating light. The induction kinetics of NPQ triggered by saturating light generally has a typical time dependence: they increase after illumination due to the initiation of electron transport and formation of NADPH preceding the activation of ATP synthase and decrease again when the Calvin cycle is activated [59,60]. In this sense, it is possible to notice the slower activation of the Calvin cycle in FS plants.

The ɸPSII is intrinsically associated with the noncyclic electron transport rates, so that the lower ɸPSII observed for *C. guianensis* in FS and *T. serratifolia* in DS affected the ETR and, consequently, the photosynthetic rate in these environments. This parameter measures the proportion of light absorbed by the PSII-associated chlorophylls that are effectively used in photochemical processes, and, as well as the F_v_/F_m_, can be used as an indicator of plant performance under different types of stress [61,62]. 

The relative contributions of the photochemical (ɸPSII) and nonphotochemical (ɸNPQ) processes for absorbed energy processing were quite divergent environments. For *B. grossularioides* in FS and *H. courbaril* and *T. serratifolia* in MS, the thermal dissipation was required in irradiance superior to the other species, corroborating with the best photochemical and photosynthetic performance of these species in the respective environments. Higher ɸPSII contributions compared to ɸNPQ indicate higher photochemical dissipation capacity without dependence on thermal dissipation, which is interesting for the studied species, since NPQ slightly differed among environments. Thus, efficient performances of photochemical extinction mechanisms are fundamental to dissipate excess energy and prevent damage. In this sense, it is emphasized that under low O_2_ conditions, in which photorespiration is suppressed, ɸNPQ surpassed ɸPSII in lower irradiance in all species and environments, independent of successional groups, evidencing that the absence of this process overloads the photochemical processes and renders the plants more susceptible to oxidative damage, which reinforces, once again, the importance of this process in the dissipation of excess energy.

### 4.3. Leaf-Nutrient Content

The inadequate supply of mineral nutrients can lead to imbalances in the photosynthetic apparatus, with reflections in chlorophyll fluorescence parameters [63]. Regarding acclimation to high or low irradiance environments, the full expression of phenotypic plasticity of species, especially of late succession, is dependent on other primary resource availability, including nutrients [47].

In general, there is great variation in the capacity of different species to extract nutrients from soil [64], which makes the concentration of leaf nutrients found in different light environments an inherent characteristic of each species [65]. The leaf-nutrient content found in this study are within the range observed in tropical tree species, despite the variation among environments [18,65,66,67].

Plants with higher growth rates, such as those exposed to full sunlight, generally require greater absorption of nutrients compared to those that grow slowly. In this study, the absorption of nutrients in DS does not seem to have been penalized, showing higher values than those observed for FS, especially for N, K, and Fe.

### 4.4. Enzymatic Activities, Leaf Phenolic Compounds, and Lipid Peroxidation

Physiological stresses can lead to disturbances in plant metabolism, which increase the production of reactive oxygen species (ROS) and may cause oxidative damage [68,69]. Plant tolerance to stress factors is associated with its antioxidant capacity and increasing levels of antioxidant constituents can prevent stress damage. Thus, an efficient antioxidant system is fundamental for the protection of the photosynthetic apparatus under conditions of stress generated by high irradiance [70].

In a study comparing the antioxidant system of tree species in different light levels, [5] concludes that the best acclimation of pioneer species to high irradiance environments is in part due to higher antioxidant enzymatic activity of this group in comparison to nonpioneer species. Contrasting with these results, no clear pattern was observed among successional groups nor greater antioxidant enzymatic activity of the two pioneer species studied.

The CAT, APX, and POX belong to different classes of H_2_O_2_ dissipation enzymes, however CAT is indicated as responsible for more efficient elimination of this compound [14,71]. In general, higher CAT activities were found in higher irradiance environments, except for *C. guianensis*, which did not differ in FS and DS (36 and 32 µmol min^−1^ mg^−1^ of protein, respectively). The opposite was observed for APX, where *H. courbaril* and *C. guianensis* had higher activity in DS (560 and 451 µmol min^−1^ mg^−1^ of protein, respectively). The enzyme CAT has a higher occurrence in peroxisomes due to the photorespiratory process, and APX occurs mainly in chloroplasts [72]. Due to the higher H_2_O_2_ formation as a consequence of photorespiration, higher CAT activity may be expected in environments with higher photorespiratory rates, in this case in FS, and higher APX activity under low irradiance (DS), where H_2_O_2_ is formed in a smaller amount [73].

The modification of the balance of enzyme activity in stressful situations can lead to compensatory mechanisms: suppressing the activity of one enzyme might induce the synthesis of another with the same purpose [74]. Although the role of APX in the elimination of H_2_O_2_ is recognized, CAT is considered more effective, which makes their greater activity in FS necessary [73]. Despite the lower CAT activity of pioneer species, *B. grossularioides* and *O. pyramidale* had higher proportional activity of CAT in relation to APX in FS (17 and four times, respectively). That demonstrates a high capacity to modulate the performance of these enzymes in function of the irradiance environment, contributing to the integrity (structural and functional) of the photosynthetic machinery.

The POX, as well as CAT and APX, play an important role in the antioxidative system detoxifying the H_2_O_2_ formed by SOD by dismutation of superoxide; however, POX had less activity than the two previous ones. Both APX and POX have several isoforms and are present in several compartments [71]. While CAT catalyzes the direct reduction of H_2_O_2_ to H_2_O and O_2_, APX eliminates H_2_O_2_ at the expense of ascorbate and POXs, thus reducing phenols. In this regard, it is believed that the main importance of POX is the oxidation of several substrates in the presence of H_2_O_2_ that could become reactive [75]. Similar to APX, the enzyme POX had less activity among the pioneers.

Regarding SOD, while [5,76] found higher SOD activity in plants of pioneer species and full sunlight acclimated leaves, in this study the results did not present a very clear pattern. Both late-successional species and the pioneer *B. grossularioides* exhibited lower SOD activity in FS and higher activity in DS, the intermediates presented lower activity in DS, and the pioneer *O. pyramidale* did not differ between FS and MS.

The SOD enzyme is considered the first line of defense in the fight against ROS, transmuting O_2_●^−^ to form H_2_O_2_, since it is the complete detoxification of the free radicals complemented by the CAT, APX, and POX enzymes. Although it is ubiquitous in aerobic organisms and subcellular compartments prone to oxidative stress, SOD relative abundance varies greatly among plants [14,15,77].

Phenolic compounds are also part of the antioxidant system [71], especially absorbing UV radiation and reducing damages caused by high irradiance. However, in this study, they had no significant participation in the prevention of cellular damage. The lipid peroxidation was estimated by the measurement of the formation of reactive substances to thiobarbituric acid (TBARS), and it was observed that the highest content of TBARS was found in plants in higher irradiance environments. The formation of TBARS in these plants is comparable to other species studied under stress conditions [77,78,79]. It is important to note that, despite the indications for lipid peroxidation, the functional stability of PSII was maintained for all plants except for *C. guianensis* in FS, as verified by the F_v_/F_m_ ratio, so the TBARS content should not be taken solely for assessment of the effectiveness of the antioxidative system.

### 4.5. Plasticity Index

High plasticity has often been reported for pioneer species, especially regarding acclimation to high irradiance, since species of this successional group generally colonize environments with greater variation in environmental conditions when compared to late-successional species [47,80,81].

The plasticity indexes obtained in this study order the species as follows: *H. coubaril* > *H. brasiliensis* > *B. grossularioides* > *T. serratifolia* > *C. guianensis* > *O. pyramidale*. The species with less plasticity belong to different successional groups, and *O. pyramidale*, a pioneer species, exhibited lower plasticity and was the only one that did not survive in all light environments. In turn, *C. guianensis* (late-successional) was the only one to undergo photoinhibition. 

Thus, our results show that the physiological plasticity of tree species in terms of photochemical/photoprotective and nutritional parameters contradicts expectations regarding the greater plasticity of pioneer species or similar responses between species of the same successional classification.

### 4.6. Relationships between Light Capture and USE Parameters, Antioxidant Activity, and Leaf-Nutrient Content

In general, in all three environments, the N content was positively related to pigment content. The positive correlation of N with the enzymatic activities in FS demonstrates the importance of the antioxidant system for the control of ROS and cellular damage, especially by the activity of SOD and by the action of carotenoids. The activity of SOD in FS was positively related to the Fe content; a component of the prosthetic group in the chloroplasts [78].

The positive correlations between P content and the photochemical processes (F_v_/F_m_, ETR, and PSII) related to efficiency in energy capture and transfer were notably higher in FS due to the higher occurrence of these events in the environment with higher irradiance. The proper functioning of light-energy conversion is an indication of the adequate supply of this nutrient in the leaves [63]. It is important to emphasize the participation of the micronutrients Mn and Fe in the increase of F_v_′/F_m_′ and F_v_/F_m_, respectively. The first has fundamental participation in the complex evolution of oxygen and the second, although acting more clearly in the electron transport chain via its metallic nature, can help to relieve the excess energy and participate effectively in the redox systems or redox signaling in the plant cells.

Positioning on the negative side of axis 1 of PCA in FS demonstrates the best photochemical performance of the pioneer species in this environment, evidenced by the higher ɸPSII, ETR, and qL. The lower antioxidant enzyme activity in *B. grossularioides* appears to have been compensated for by phenolic compounds but is still insufficient to reduce oxidative damage (TBARS). The lower damage in *O. pyramidale* (14.5 µmol g^−1^ DM) seems to be a consequence of the greater activity of SOD (212.2 Unit mg^−1^ of protein).

With an irradiance decrease in MS and DS, the effect of SOD activity in reducing TBARS also decreases. This result suggests a convergence (SOD activity and lipid peroxidation) at lower energization states. In MS, the participation of P in photochemical processes decreased in relation to FS, contributing only to the increase of qL. However, the negative P–TBARS relationships indicate that this element contributed to the reduction of cellular damage, probably increasing the efficiency of reactions that require energy transfer mediated by ATP. The positive effects of Mn on fluorescence parameters were repeated in MS, with emphasis on the increase of ɸPSII=ɸNPQ, probably because this conferred greater stability to the functioning of PSII.

The positioning of *H. brasiliensis* and *B. grossularioides* in the PCA of MS suggests that the lower oxidative damages observed in these species are due to the higher antioxidant enzymatic activity in the first (CAT) and the high qL and NPQ for the second.

In DS, ɸPSII = ɸNPQ was also important, where, together with ETR and ɸPSII, the negative correlations with TBARS show that efficient mechanisms of photochemical extinction in plants under intense shading may be more decisive in the prevention of oxidative damage than a strong antioxidant system, since this would require an investment of part of the nitrogen allocated to capture and transfer energy [35].

PCA in DS allows us to infer that the higher ɸPSII in *B. grossularioides* and *H. brasiliensis* can be attributed, at least partially, to the higher pigment content and lower dissipations regulated and not regulated by PSII (ɸNPQ and ɸNO, respectively), as well as to the increase in P content.

## 5. Conclusions

In terms of plasticity of photochemical, nutritional, and antioxidant parameters, the species exhibited different results than usually expected in terms of successional groups, with a pioneer species exhibiting low plasticity and a late-successional being more plastic, the results presented in this study support the original hypothesis about the low level of responsiveness to species classification in the successional groups, especially because the antioxidant system presented few significant differences between groups. In addition, it was observed that the late-successional species exhibited higher phenolic content and lower SOD activity under full sunlight, diverging from the intermediate and pioneer ones.

The most efficient way of avoiding oxidative damage to plants under intense shade was the increase of photochemical quenching and the use of carotenoids for thermal dissipation via the xanthophyll cycle. In this respect, we highlight the positive effect of phosphorus on the increase of PSII yield in all environments, especially those with higher irradiance. 

Finally, the pioneer species presented better photochemical performance under high irradiance compared to nonpioneer species. The late-successional *C. guianensis* was the only one that showed signs of photoinhibition, and the other species, despite the signs of lipid peroxidation in FS, exhibited adequate PSII yield and showed themselves to be able to withstand the high irradiance. On the other hand, the death of *O. pyramidale* and the increased SOD activity in DS recorded for *B. grossularioides* suggest that the stress due to low light availability for pioneer plants may be more difficult to overcome than the high irradiance for late-successional species.

## Figures and Tables

**Figure 1 plants-09-01047-f001:**
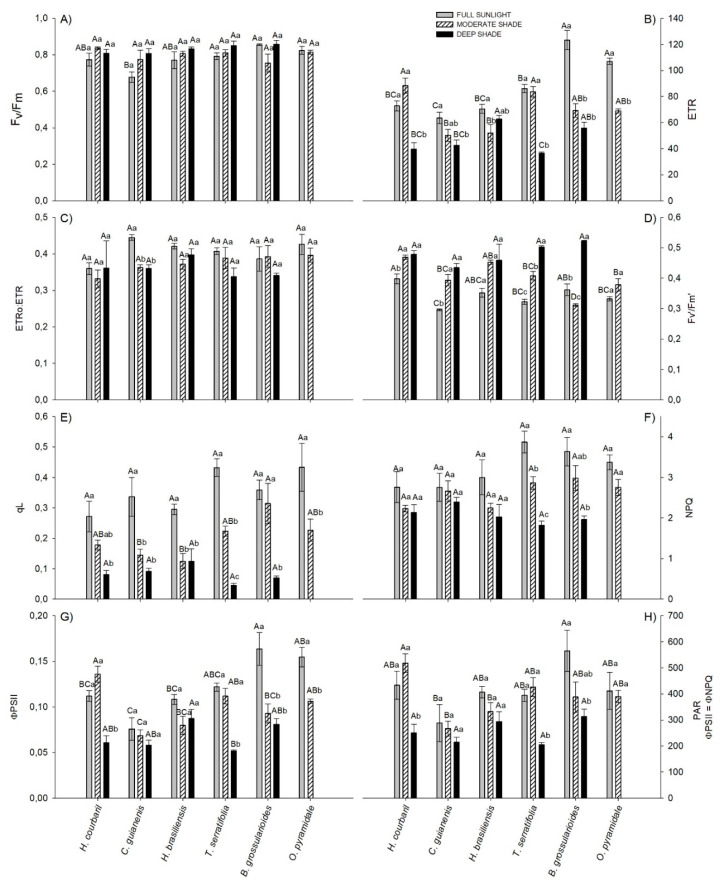
Fluorescence parameters at saturating light (PAR = 1500 µmol m^−2^ s^−1^) of six tree species submitted to three different light environments: (**A**) Maximum quantum efficiency of PSII, (**B**) Electrons transport rate, (**C**) Fraction of electrons destined for photorespiration, (**D**) Maximum efficiency of PSII in the light, (**E**) Photochemical quenching, (**F**) Non-photochemical quenching, (**G**) Effective quantum efficiency of PSII and (**H**) Irradiance wich which the non-photochemical quenching yield overcome the Effective quantum yield of PSII. Same capital letters for different species in same environment and small case for same species in different environment are equal by Tukey test (*p* < 0.05). Vertical bars indicate the standard error (*n* = 4).

**Figure 2 plants-09-01047-f002:**
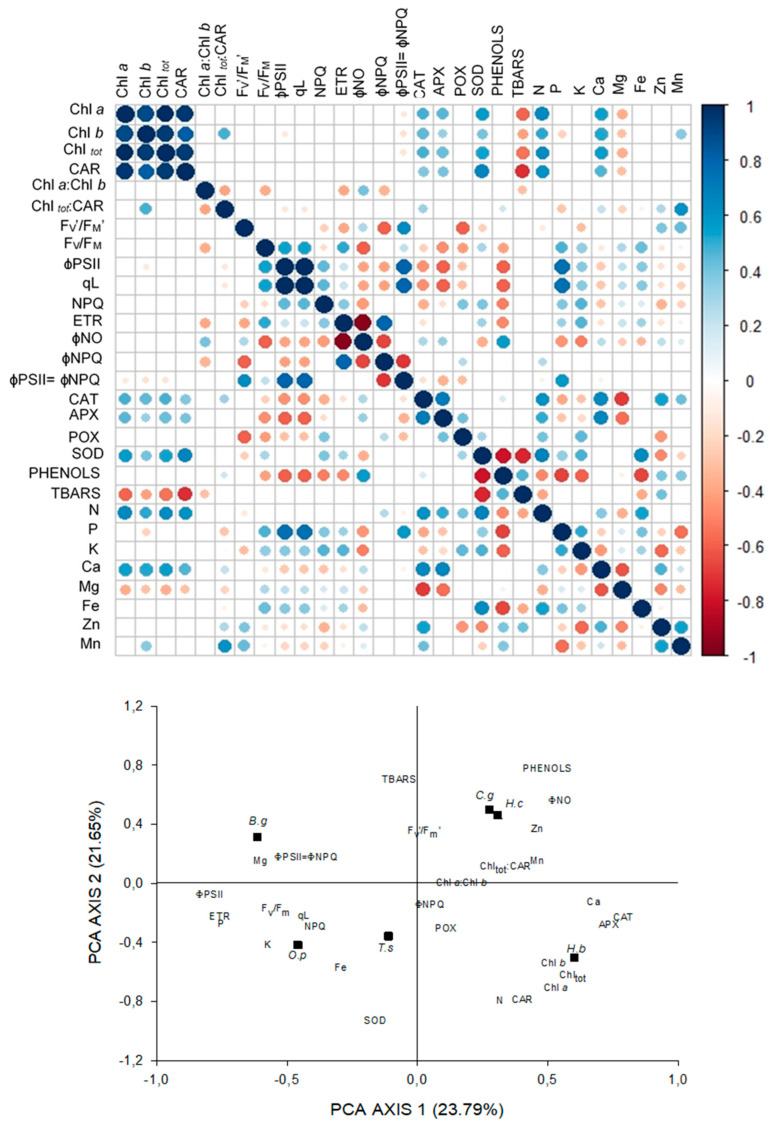
Correlations between pigments content, fluorescence parameters, antioxidant enzymatic activity, phenolic compounds, lipid peroxidation, and foliar concentration of nutrients in six tree species under full sunlight, and principal components analysis of the 29 variables studied. Black squares represent the species: *Hymenea courbaril (H.c); Carapa guianensis (C.g); Hevea brasiliensis (H.b); Tabebuia serratifolia (T.s); Bellucia grossularioides (B.g), and Ochroma pyramidale (O.p).*

**Figure 3 plants-09-01047-f003:**
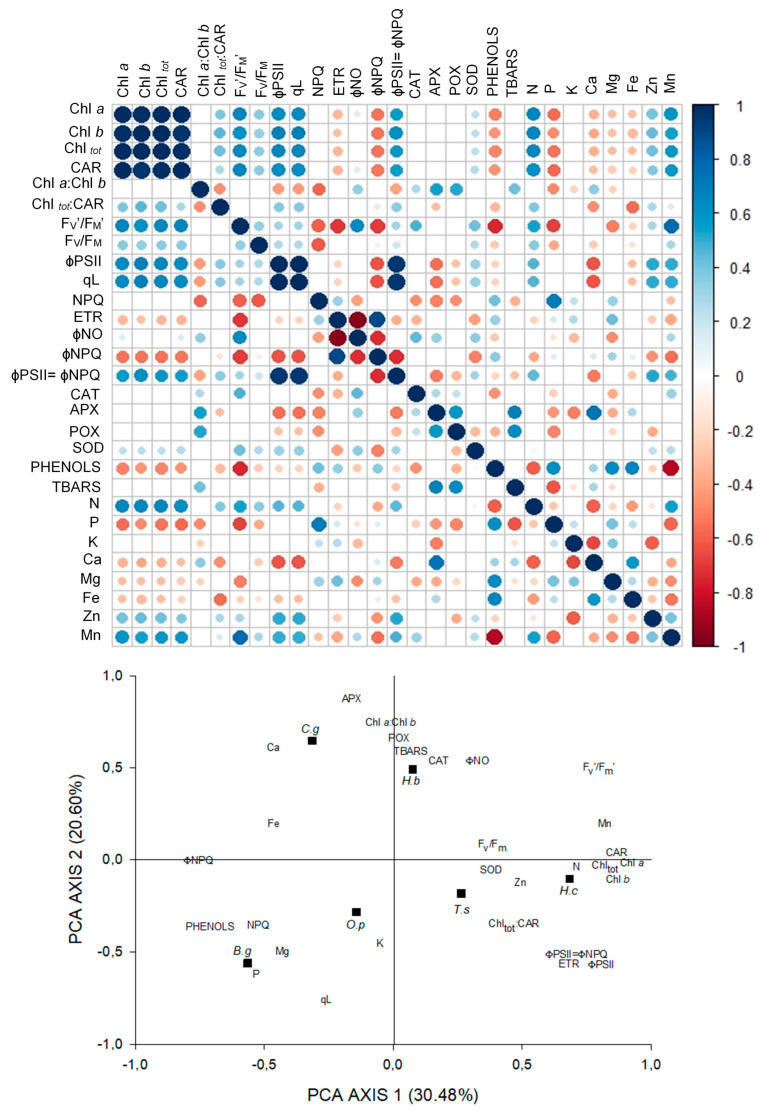
Correlations between pigments content, fluorescence parameters, antioxidant enzymatic activity, phenolic compounds, lipid peroxidation, and foliar concentration of nutrients in six tree species under moderate shade, and principal components analysis of the 29 variables studied. Black squares represent the species: *Hymenea courbaril (H.c); Carapa guianensis (C.g); Hevea brasiliensis (H.b); Tabebuia serratifolia (T.s); Bellucia grossularioides (B.g), and Ochroma pyramidale (O.p).*

**Figure 4 plants-09-01047-f004:**
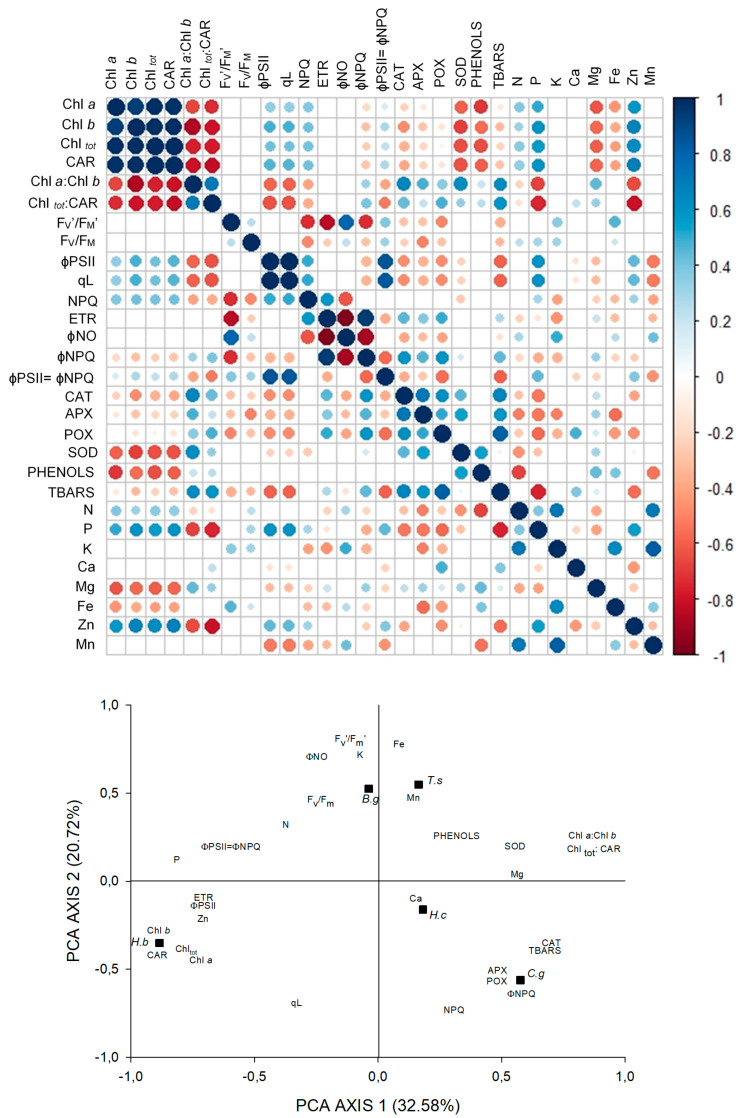
Correlations between pigments content, fluorescence parameters, antioxidant enzymatic activity, phenolic compounds, lipid peroxidation, and foliar concentration of nutrients in six tree species in deep shade, and principal components analysis of the 29 variables studied. Black squares represent the species: *Hymenea courbaril (H.c); Carapa guianensis (C.g); Hevea brasiliensis (H.b); Tabebuia serratifolia (T.s); Bellucia grossularioides (B.g), and Ochroma pyramidale (O.p).*

**Table 1 plants-09-01047-t001:** List of studied species with scientific name, family, and successional groups, as described in literature.

Specie	Family	Successional Groups
*Hymenea courbaril*	Fabaceae	Late successional [19]
*Carapa guianensis*	Meliaceae	Late successional [20]
*Hevea brasiliensis*	Euphorbiaceae	Mid-successional [21]
*Tabebuia serratifolia*	Bignoniaceae	Mid-successional [22]
*Bellucia grossularioides*	Melastomataceae	Pioneer [23]
*Ochroma pyramidale*	Malvaceae	Pioneer [24]

**Table 2 plants-09-01047-t002:** Daily PPFD (photosynthetic photon flux density) average, maximum PPFD observed, average percentage of full sunlight under three light environments.

	Full Sunlight (FS)	Moderate Shade (MS)	Deep Shade (DS)
PPFD average (µmol photons m^−2^ s^−1^)	1027.51 ± 10.49	362.50 ± 9.37	47.95 ± 7.98
Maximum PPFD (µmol photons m^−2^ s^−1^)	1866.17 (13:00 h)	795.61 (12:00 h)	74.50 (14:00 h)
% PPFD	100	35.23	4.66

**Table 3 plants-09-01047-t003:** Fluorescence parameters calculated from saturating pulse analysis.

Fluorescence Parameter	Name and Physiological Interpretation
F_v_/F_m_ = (F_m_ − F_0_)/F_m_	Estimated maximum quantum efficiency (yield) of PSII photochemistry [27]
F_v_′/F_m_’ = (F_m_′ − F_0′_)/F_m_′	Maximum efficiency of PSII photochemistry in the light, if all centers were open [28]
ɸPSII = (F_m_′ − F′)/F_m_′	Estimated effective quantum yield (efficiency) of PSII photochemistry at incident PAR [29]
ETR = αβ x PAR x ɸPSII	Electron transport rate in PSII at incident PAR. (αβ was determined from curves obtained under nonphotorespiratory conditions in an atmosphere containing less than 1% O_2_) [30,31]
NPQ = (Fm − F_m_′/F_m_′)	Non-photochemical quenching [32,33]
qP = (F_m_′ − F_s_)/(F_m_′ − F_0′_)	Coefficient of photochemical quenching based on the “puddle” model (i.e., unconnected PSII units) [34]
qL = qP x (F_0′_/F_s_)	Coefficient of photochemical quenching based on the “lake” model (i.e., fully connected PSII units) [28]
ɸNO=1/[NPQ+1+qL(F_m_/F_0_ − 1)]	Quantum yield of nonregulated energy dissipation in PSII [28]
ɸNPQ = 1 − ɸPSII − ɸNO	Quantum yield for dissipation by down-regulation [28]

**Table 4 plants-09-01047-t004:** Plasticity index of fluorescence, nutritional, and antioxidant parameters of six tree species submitted to three light environments.

	*H. courbaril*	*C. guianensis*	*H. brasiliensis*	*T. serratifolia*	*B. grossularioides*	*O. pyramidale*	Mean Values
Fluorescence Parameters							
F_V_/F_M_	0.22	0.28	0.23	0.17	0.27	0.11	0.21 ± 0.07 ^D^
ETR	0.74	0.50	0.48	0.63	0.68	0.42	0.57 ± 0.13 ^B^
ETR_O_/ETR	0.64	0.27	0.25	0.39	0.32	0.25	0.35 ± 0.15 ^CD^
F_V_′/F_M_′	0.30	0.39	0.43	0.41	0.43	0.26	0.37 ± 0.07 ^CD^
qL	0.86	0.84	0.87	0.95	0.89	0.76	0.86 ± 0.06 ^A^
NPQ	0.54	0.44	0.65	0.64	0.60	0.33	0.53 ± 0.12 ^BC^
ΦPSII	0.74	0.54	0.48	0.63	0.68	0.42	0.58 ± 0.12 ^B^
						Mean Values	0.50 ± 0.08 ^(2)^
Nutritional Parameters							
N	0.60	0.49	0.61	0.61	0.63	0.41	0.56 ± 0.09 ^A^
P	0.55	0.46	0.68	0.50	0.82	0.62	0.61 ± 0.13 ^A^
K	0.83	0.75	0.84	0.72	0.73	0.73	0.77 ± 0.06 ^A^
Ca	0.65	0.54	0.74	0.84	0.54	0.35	0.61 ± 0.17 ^A^
Mg	0.78	0.77	0.83	0.69	0.43	0.53	0.67 ± 0.16 ^A^
Fe	0.59	0.76	0.38	0.79	0.77	0.35	0.61 ± 0.20 ^A^
Mn	0.84	0.78	0.49	0.74	0.74	0.75	0.72 ± 0.12 ^A^
Zn	0.65	0.49	0.88	0.61	0.53	0.46	0.60 ± 0.15 ^A^
N:P	0.48	0.61	0.76	0.66	0.91	0.59	0.67 ± 0.15 ^A^
						Mean Values	0.65 ± 0.06 ^(1,2)^
Antioxidant Parameters							
CAT	0.83	0.75	0.90	0.71	0.88	0.20	0.71 ± 0.26 ^A^
APX	0.99	0.90	0.79	0.33	0.93	0.57	0.75 ± 0.25 ^A^
POX	0.99	0.75	0.83	0.49	0.70	0.72	0.75 ± 0.17 ^A^
SOD	0.87	0.75	0.95	0.64	0.71	0.43	0.73 ± 0.18 ^A^
PHENOLS	0.62	0.57	0.71	0.63	0.57	0.67	0.63 ± 0.06 ^A^
						Mean Values	0.71 ± 0.14 ^(1)^
Mean Values	0.68± 0.20 ^a^	0.60±0.18 ^ab^	0.66 ± 0.22 ^a^	0.61 ± 0.18 ^ab^	0.66 ± 0.19 ^a^	0.47 ± 0.19 ^b^	

The different lowercase letters denote significant differences between the mean values for species, the different uppercase letters denote significant differences among variables in the same parameter, and the number denotes differences among the fluorescence, nutritional, and antioxidant parameters.

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
