# Peer review of "Photochemical Efficiency and Oxidative Metabolism of Tree Species during Acclimation to High and Low Irradiance"

_plants, 2020, doi:10.3390/plants9081047_

Round 1

Reviewer 1 Report

The manuscript compares sensibility to photoinhibition and their photosynthetic performances of tropical tree species from different ecological groups, (pioneer-, middle-, and late- successional groups).  

The manuscript is well written, is well structured, respect scientific formats for its tables and figures and is easy to read. Detailed analysis of chlorophyll fluorescence parameters and photosynthesis is well conduced. Authors should add how they determine the saturating light intensity for each species at each light acclimation treatment, and if they used the same saturating light for fluorimetric study thereafter.

Protocols of enzyme activity are not provided. They must be described because readers need of these information for data interpretation. How many grams of fresh leaves they used? When leaves have been collected (early before photosynthesis, at midday, at night?),…           

 Plasticity indexes are proposed to analyze tree acclimation response to light. Plasticity indexes of variables such as fluorescence parameters and mineral contents can be used in interspecies comparisons. Precaution must be carried out in interpretation of plasticity index calculated for the global enzyme activities. Despite the antioxidants and the precautions used, the grinding of the leaves is sufficient to oxidize (instantaneously) the enzymes of some species while this does not occur for other tree species. Finally, comparisons of plasticity index of phenolics based on Folin-Ciocalteau analyses are not relevant. “Total phenolic compounds” determined by using this reagent only concerned extractible phenolic compounds with ethanol or equivalent. These trees have phenolic compounds of different subclasses, which react differently to the reagent. Furthermore, it is well known that full sun leaves esterify their phenolic compounds within cell walls, polymerize larger amount of lignin within the blade. These compounds are not extractible with protocol adopted for Folin-Ciocalteau, and are not taken into account. It is also well established that shade leaves accumulate large amounts of hydroxycinnamic acids which are replaced by flavonoids in full sun light. These compounds do not react with comparable intensity at the Folin Ciocolteux reagent. Hevea brasiliensis, Hymenea courbaril, Bellucia grossularioides have totally different extractible phenolics and tannins. Consequently, the interspecies comparisons without detail analyses of phenolic compounds are not relevant.     

Analyze of leaf mineral contents need to be associated to the vegetative growth rate, because growing plants exhibited lower content in certain minerals.

The referee recommends a minor revision of the manuscript to improve these four points.

Author Response

REVIEWER 1 (Associate Editor’s Comments)

1-The manuscript is well written, is well structured, respect scientific formats for its tables and figures and is easy to read. Detailed analysis of chlorophyll fluorescence parameters and photosynthesis is well conduced. Authors should add how they determine the saturating light intensity for each species at each light acclimation treatment, and if they used the same saturating light for fluorimetric study thereafter.

Authors: The authors would like to thank you for your kind words regarding the manuscript. The saturation irradiance was determined by means of previously performed light curves (manuscript in preparation) and the same intensity of actinic light (1500 µmol m-2 s-1) and the saturating pulse (8000 µmol m-2 s-1) to measure fluorescence was used for all plants and treatments. The information was inserted in text between lines 143 and 149.

2-Protocols of enzyme activity are not provided. They must be described because readers need of these information for data interpretation. How many grams of fresh leaves they used? When leaves have been collected (early before photosynthesis, at midday, at night?)

Authors: The suggestions have been accepted.

All samples were collected at 1:00 pm, as described at lines 158-159. For the analysis of enzyme activity, 200 mg samples from leaves were used in these analyzes. This information was added to line 159. The extraction procedures are detailed among lines 161-163, including centrifugation, time and temperature. After extraction, the activity of each enzyme was quantified by a specific protocol. The basic information and references are cited between lines 168 and 171.

3-Plasticity indexes are proposed to analyze tree acclimation response to light. Plasticity indexes of variables such as fluorescence parameters and mineral contents can be used in interspecies comparisons. Precaution must be carried out in interpretation of plasticity index calculated for the global enzyme activities. Despite the antioxidants and the precautions used, the grinding of the leaves is sufficient to oxidize (instantaneously) the enzymes of some species while this does not occur for other tree species. Finally, comparisons of plasticity index of phenolics based on Folin-Ciocalteau analyses are not relevant. “Total phenolic compounds” determined by using this reagent only concerned extractible phenolic compounds with ethanol or equivalent. These trees have phenolic compounds of different subclasses, which react differently to the reagent. Furthermore, it is well known that full sun leaves esterify their phenolic compounds within cell walls, polymerize larger amount of lignin within the blade. These compounds are not extractible with protocol adopted for Folin-Ciocalteau, and are not taken into account. It is also well established that shade leaves accumulate large amounts of hydroxycinnamic acids which are replaced by flavonoids in full sun light. These compounds do not react with comparable intensity at the Folin Ciocolteux reagent. Hevea brasiliensis, Hymenea courbaril, Bellucia grossularioides have totally different extractible phenolics and tannins. Consequently, the interspecies comparisons without detail analyses of phenolic compounds are not relevant.

Authors: We appreciate your considerations. The extraction of phenolic compounds followed the protocols described in the literature and precautions were taken to avoid oxidation. Differences between species and environments must be considered to choose methods and interpret results, but this is exactly the aim of plasticity studies: to evaluate responses to different conditions. We are sure that our results were obtained following the protocols in a reliable manner and already established in the literature for the evaluation of phenolic compounds in plants under different growing conditions (see Tiepo, A.N., Constantino, L.V., Madeira, T.B. et al. (2020). Plant growth-promoting bacteria improve leaf antioxidant metabolism of drought-stressed Neotropical trees. Planta 251, 83; Chua, I. Y. P, King, P. J. H, Ong, K. H, Sarbini, S. R, & Yiu, P. H. (2015). Influence of light intensity and temperature on antioxidant activity in Premna serratifolia L. Journal of soil science and plant nutrition, 15(3), 605-614; Tai-Jie Zhang, Jin Zheng, Zheng-Chao Yu, Xuan-Dong Huang, Qi-Lei Zhang, Xing-Shan Tian, Chang-Lian Peng. (2018). Functional characteristics of phenolic compounds accumulated in young leaves of two subtropical forest tree species of different successional stages, Tree Physiology, 38 (10), 1486–1501; Yu, Z.; Zheng, X.; Lin, W.; Cai, M.; Zhang, Q.; Peng, C. 2020. Different photoprotection strategies for mid- and late-successional dominant tree species in a high-light environment in summer. Environ. Exp. Bot., 171, 103927).

Thus, we believe that the data presented in this study can contribute to the understanding of the photoprotective role of enzymatic and nonenzymatic antioxidant systems during the acclimatization of tropical tree species seedlings to different light conditions.

4-Analyze of leaf mineral contents need to be associated to the vegetative growth rate, because growing plants exhibited lower content in certain minerals.

Authors: The leaf nutrient content certainly has a close relationship with growth rates, but in this study, we aimed to relate the content of nutrients with photochemical performance and the antioxidant system in each environment. Taking into account that the seedlings ages were similar and in this vegetative phase they were grown on the same substrate, the differences in the leaf nutrient content can be attributed to changes in metabolism induced by difference in the light environments. In addition, assessing the nutritional status of the plants in each environment (supplementary files), as well as the photochemical performance, there is no evidence of mineral deficiency.

Reviewer 2 Report

Abstract: Please write out full words when first using (eg. SOD and LHC)

Line 28: the use of “in opposite to expected” is awkward, please re-phrase.

Introduction: the first paragraph is somewhat choppy and should be re-written to be easier to read.

Line 49: The use of ontogenetic stages is confusing. The field of ontology is typically referring to the development of an organism from fertilization to adult. Specifically, it relates to humans and mammals. Also, it does not typically account for the variation in environment. I suggest using a different word.

Line 60-63: This needs to be re-written as they do not make sense to the reader and is a run-on sentence.

Line 67: Please write out SOD in full first.

Line 68: Please write out CAT, APX, and POX as well.

Line 73: There is a missing period after the references.

Introduction: Overall, I think the discussion is somewhat shallow with respect to related literature. I think the authors should present a better background related to their topic. Also, I find the introduction hard to read. The English and sentence structure should be improved.

Table 2: Please write out PAR first. Also indicate what units are used (micromole/m2/s)

Line 137: Please indicate the lighting source used. Some Li-COR fluorometer heads only produce red and blue light.

Line 141: Indicate lighting source used and size of the chamber.

Line 146-147: Authors indicate that the lighting levels were used in decreasing order yet write them in ascending order. Please correct this so they both indicate the same.

Table 3: I suggest adding an addition space between each row. There are places where it is difficult to identify which definition is related to which parameter (i.e., ETR).

2.8 Plasticity index: In the equation the authors using maximum mean value and minimum mean value. Please describe these terms more. What is the maximum and minimum values? Which parameters are being used?

Line 216: Without a reference to what a maximum Fv/Fm ratio is, the value of 0.75 is useless to readers. Please give a reference stating maximum value (0.82; Baker, 2008) is typically used.

Figure 1: Please ensure all statistical letter groups are able to be seen and do not interfere with other bars.

Table 4: There are superscript numbers in each mean value column. What do those relate to? Also, on the last light of the table the furthest mean to the right, there needs to be a super script.

Line 366: Replace the intensity with rate.

Line 401: It should be PSII not FSII.

Overall Discussion: While it is well written, similar to the introduction, the English and sentence structure need to be improved. There are a lot of run-on sentences making it harder than it has to be to read. I encourage the authors to keep only one idea within each sentence. Furthermore, I feel the discussion lacks the implications of the finds and how they help each succession or species survive in each habitat. I feel that the authors have done an extensive study of a broad range of species and should talk about this. To me the discussion seems to mimic a lot of the results, re-stating points. Please make the discussion more direct and include implications of this research.

Author Response

REVIEWER 2 (Reviewer’s Comment)

1.Abstract:

Please write out full words when first using (eg. SOD and LHC)

Line 28: the use of “in opposite to expected” is awkward, please re-phrase.

Authors: Yes, we agree. The full words were included (lines 23 and 26) and we re-write the phrase (lines 27-29).

2.Introduction:

Overall, I think the discussion is somewhat shallow with respect to related literature. I think the authors should present a better background related to their topic. Also, I find the introduction hard to read. The English and sentence structure should be improved.

Authors: Yes, improvements have been made in the introduction. Edits had been performed for the English language, clarity and fluidity of the text.

Authors: The first paragraph is somewhat choppy and should be re-written to be easier to read.

Authors: Yes, the suggestion was accepted.

Line 49: The use of ontogenetic stages is confusing. The field of ontology is typically referring to the development of an organism from fertilization to adult. Specifically, it relates to humans and mammals. Also, it does not typically account for the variation in environment. I suggest using a different word.

Authors: Yes, it was done (lines 44-49).

Line 60-63: This needs to be re-written as they do not make sense to the reader and is a run-on sentence

Authors: The phrase has been rewritten (lines 57-63).

Line 67: Please write out SOD in full first.

Authors: The full words included (lines 65-67).

Line 68: Please write out CAT, APX, and POX as well.

Authors: Yes, it was done.

Line 73: There is a missing period after the references.

Authors: Thank you for your comment. The period was inserted (line 73).

Authors: In general, several enhancements have been made to the introduction as a whole. So, we hope that our edits meet your expectations.

3. Material and methods

Table 2: Please write out PAR first. Also indicate what units are used (micromole/m2/s)

Authors: Yes, the suggestion was accepted (line 105). We have opted to replace PAR to PPFD (photosynthetic photon flux density).

Line 137: Please indicate the lighting source used. Some Li-COR fluorometer heads only produce red and blue light.

Authors: We made improvements in the text (line 134). The leaf chamber with attached light source used was the model 6400-40, with 2cm2 of measurement area.

Line 141: Indicate lighting source used and size of the chamber.

Authors: Yes, it was done (lines 143 to 149).

Line 146-147: Authors indicate that the lighting levels were used in decreasing order yet write them in ascending order. Please correct this so they both indicate the same.

Authors: Yes, it was done.

Table 3: I suggest adding an addition space between each row. There are places where it is difficult to identify which definition is related to which parameter (i.e., ETR).

Authors: The suggestion was accepted. Space has been added between paragraphs.

2.8 Plasticity index: In the equation the authors using maximum mean value and minimum mean value. Please describe these terms more. What is the maximum and minimum values? Which parameters are being used?

Authors: We made improvements in the text (lines 191-193). The plasticity index (PI) is calculated for species and variables using the difference between the maximum and minimum values for each variable studied, regardless of treatments. It seeks to demonstrate the response plasticity of a given species to different environmental conditions for the study parameter. In this case, the parameters studied are displayed in Table 4. In addition, this index is widely studied and usually presented as we have done it here (please, see Valladares F, Niinemets U (2008) Shade tolerance, a key plant feature of complex nature and consequences. Annu Rev Ecol Evol Syst 39:237–257; Marenco, R.A., Camargo, M.A.B., Antezana-Vera, S.A. et al. (2017) Leaf trait plasticity in six forest tree species of central Amazonia. Photosynthetica 55, 679–688 and dos Santos, O. de O.; Mendes, K.R.; Martins, S.V.C.; Batista-Silva, W.; dos Santos, M.A.; de Figueirôa, J.M.; de Souza, E.R.; Fernandes, D.; Araújo, W.L.; Pompelli, M.F. Physiological parameters and plasticity as key factors to understand pioneer and late successional species in the Atlantic Rainforest. Acta Physiol. Plant. 2019, 41, 1–18, doi:10.1007/s11738-019-2931-9

4. Results:

Line 216: Without a reference to what a maximum Fv/Fm ratio is, the value of 0.75 is useless to readers. Please give a reference stating maximum value (0.82; Baker, 2008) is typically used.

Authors: Yes, we agree. This point was clarified in the discussion, where we include the references (line 399).

Figure 1: Please ensure all statistical letter groups are able to be seen and do not interfere with other bars.

Authors: Yes, all statistical letter groups are visible and without interfering with the others.

Table 4: There are superscript numbers in each mean value column. What do those relate to? Also, on the last light of the table the furthest mean to the right, there needs to be a super script.

Authors: Yes, we agree. These comments were added in bottom of Table 4:

The different lowercase letters denote significant differences between the mean values for species, the different uppercase letters denote significant differences among variables of the same parameter and the number denote differences among the fluorescence, nutritional and antioxidant parameters.

Line 366: Replace the intensity with rate

Authors: Yes, it was done. The intensity was replaced to rate (line 380).

Line 401: It should be PSII not FSII

Authors: Yes, that was fixed. PSII is the correct.

5. Overall Discussion: While it is well written, similar to the introduction, the English and sentence structure need to be improved. There are a lot of run-on sentences making it harder than it has to be to read. I encourage the authors to keep only one idea within each sentence. Furthermore, I feel the discussion lacks the implications of the finds and how they help each succession or species survive in each habitat. I feel that the authors have done an extensive study of a broad range of species and should talk about this. To me the discussion seems to mimic a lot of the results, re-stating points. Please make the discussion more direct and include implications of this research.

Authors: We appreciate your comments. The text underwent a structural and grammatical revision by a native speaker in English, so we believe that now the text will be in a better condition to meet the publication criteria. As for the discussion of the results, it was conducted in line with the objectives of this work, which aims to investigate the photochemical performance and the behavior of the antioxidant system between species of different functional groups and understand how both contribute to avoid photoinhibitory damage, due to changes in the light environment. In this sense, although the ecological aspects are very relevant for the discussion, we believe that they have the focus and the necessary attention for the scope of the study. In that way, we hope to have clarified all outstanding issues.
